# Screwed Monolithic Zirconia Crowns for Mono-Implant Posterior Rehabilitation: A Prospective Clinical Study on 41 Patients with a 7-Year Follow-Up

Giuseppe Barile [1],[*],[†], Saverio Capodiferro [1],[*],[†], Giovanni De Rosa [1], Giovannino Muci [2], Alessandro Vanzanelli [2] and Massimo Corsalini [1]

1   Department of Interdisciplinary Medicine, 'Aldo Moro', University of Bari, 70100 Bari, Italy; giovanni.derosa60@gmail.com (G.D.R.); massimo.corsalini@uniba.it (M.C.)
2   Department of Prosthetic Dentistry, Dental School of Bari, University of Bari, 70100 Bari, Italy; giovanninomuci@studiomuci.it (G.M.); alexvanzanelli@libero.it (A.V.)
*   Correspondence: giuseppe.barile@uniba.it (G.B.); saverio.capodiferro@uniba.it (S.C.)
†   These authors contributed equally to this work.

**Abstract:** The usage of monolithic zirconia has been increasing in daily practice in dentistry for the last 20 years. Monolithic zirconia is mainly used for posterior sector rehabilitation, as it lacks optical properties and has good technical properties; it does not chip and reduces antagonist wear compared to metal–ceramic prostheses. However, monolithic zirconia may present some issues, mainly low-temperature degradation (LTD), also known as "aging", which must be considered and investigated in clinical studies with prolonged follow-up periods. The aim of this study was the aesthetic and functional evaluation of single monolithic zirconia crowns that were screwed onto implants for posterior dental rehabilitation using USPHS parameters and a long follow-up period of 7 years. The results showed that the USPHS scoring reduces year by year, passing from 100% excellence between the first and fourth years of observation, to 88% excellence and 12% success in the seventh year. Screw abutment loosening was the only technical adversity reported; no implant failure, crown fracture, or irreparable damage were additionally registered. Considering the emerging results and comparing them with the data available in the literature, the authors suggest that monolithic zirconia crowns could be considered as a valid alternative to classic metal–ceramic rehabilitations for the rehabilitation of posterior sectors.

**Keywords:** monolithic zirconia; implant; dental prosthesis; single crown; USPHS

## 1. Introduction

Currently, conventional metal–ceramic prostheses still represent the gold standard in prosthetic restorative materials due to their excellent long-term mechanical properties, especially for their use as fixed dental prostheses (FDPs) mainly in the posterior sectors [1].

An alternative solution that aims to progressively substitute metal–ceramic prostheses involves the use of zirconia. Zirconia's aesthetic and mechanical properties are improving year by year, and are becoming more predictable, popular, and capable of replacing metal–ceramics in almost any application.

The name "Zirconia" refers to zirconium dioxide ($ZrO_2$); zirconium (Zr) is a common metal element of the Earth's crust, being seventeenth on the relative abundance scale, greater than copper, tin, lead, or zinc; its high availability makes the cost of the material cheaper than others that are currently used for fixed prostheses [2].

The most-used prosthetic dental crowns are tetragonal zirconia polycrystals (TZP) and partially stabilized zirconia (PSZ) forms; TZP, more commonly Y-TZP, has the highest values for flexural strength, and PSZ could be composed of an increasing %mol of yttria, which allows for improved mechanical properties [3]. Following these considerations,

zirconia was classified by Zhang and Lawn in three generations based on the mol% of yttria that is used to stabilize its crystalline form (upon which depends its mechanical properties, such as elasticity modulus, flexural strength, and fracture toughness); precisely, the first two generations are composed of 3Y-TZP, while the third is composed of 4Y-PSZ and 5Y-PSZ [2].

Despite its good aesthetic and clinical performance, zirconia presents some adverse effects. The most common is the spontaneous degradation of zirconia related to the metastability of the tetragonal phase, which could be transformed into a monoclinic phase if it is triggered by certain factors, such as mechanical contributions or thermal energy, or through chemical reactions on the surface [4]. In fact, Kobayashi et al., in 1981, observed that the slow and progressive spontaneous transformation of the tetragonal phase to monoclinic is associated with a reduction in the material's mechanical properties over time [5]. It is believed that such a phenomenon, known as "aging" or "low-temperature degradation" (LTD), was related to water reacting with the crystal matrix. This transformation causes the formation of surface micro-cracks, facilitating water penetration followed by propagation in the framework volume.

Adjunctive adversities that are generally reported are chipping and cohesive bond loss (CBL). Chipping is defined as feldspathic ceramic veneering fractures, due to them having a lower strength than zirconia cores, and CBL is the lack of adhesion between the veneer and the core [6]. The chipping phenomenon belongs exclusively to veneered zirconia, which differs from monolithic zirconia due to the absence of ceramic veneering in the latter. In addition to this essential technical advantage, monolithic zirconia presents some optical problems due to its reduced translucency: the first generations were indicated only for the posterior sectors, but the latest solutions offered by manufacturing companies have overcome this optical issue, providing clinicians with monolithic zirconia, which could also be used in anterior sectors [7].

Monolithic zirconia has already been used for a long time as a prosthetic crown screwed onto an implant, as reported by several authors, with overlapping results. In one of the most recent systematic reviews on this topic, Kim et al. considered many studies evaluating the clinical outcomes of monolithic zirconia supported by an implant, both as a single crown and as an FDP [8]; the longest follow-up was reported in two studies that provided five years of observation and reported a 99.84% cumulative survival rate [9,10]. In a systematic review with a meta-analysis by Pjetursson, monolithic zirconia crowns screwed onto implants provide a 96.1% survival rate at three years of follow-up, similar to veneered restorations, even if the failure rate of the latter is higher than the monolithic form [11]. The most frequent technical adversity that has been reported is screw loosening, which causes a loss of retention that is higher than that with veneered crowns [12]. However, unlike chipping and feldspathic fractures, screw loosening is always considered to be a reparable technical adversity, addressable with the tightening of the screw. The only irreparable technical adversity shown with monolithic zirconia is the fracture of the implant abutment due to an excessive masticatory load, as described by Laumbacher et al. [13].

Despite many studies which have exclusively considered the survival of this kind of restoration and described its excellent technical outcomes, there is a lack of long-term RCT studies with clinical evaluations. In fact, the latest study that evaluates clinical performances was conducted in 2022 by Gao et al., and reported a comparison between conventional and digital workflows evaluated based on the USPHS criteria [14]. The United States Public Health Service (USPHS) criteria are the most used criteria to perform a standard evaluation of the clinical outcomes of dental restoration [15], and provide a score from alpha (excellence restoration) to delta (restoration that must be replaced) regarding marginal adaption, surface roughness, color matching, anatomic morphology, and fracture. Zarone et al. used the USPHS criteria to evaluate the clinical performance of three units of zirconia FDPs over fourteen years, and reported that only 2% of the FDPs needed to be replaced after irreparable FDP fractures [16]. Moreover, the Italian Academy of Prosthetic

Dentistry (AIOP) reported the same percentage of delta USPHS scores on 149 monolithic single crowns due to irreparable fractures [17].

Considering all of the aforementioned aspects and the need in the literature to have a clinical evaluation through standard parameters with a long-term follow-up period, the current prospective study aims to evaluate the qualitative status of implant-supported screw-retained monolithic single crowns (SC) throughout a period of 7 years, annually followed up each time with USPHS scoring for the entire observational period.

## 2. Materials and Methods

### 2.1. Patient Sampling

The authors considered a convenience sample of patients with single edentulous spaces of the posterior sectors, maxillary or mandibular, with adjacent teeth lacking periapical or periodontal lesions, who needed fixed implant prosthetic rehabilitation. The study was conducted in accordance with the Declaration of Helsinki, and the protocol was approved by the Ethical Committee of University Hospital "Policlinico di Bari" (Study No. 7390, N. Prot. 0069684).

As provided in the specific section of the Italian Code of Medical Ethics 'Information and Communication. Consent and dissent—Title VI', each patient received understandable and comprehensive information on the techniques and materials, diagnostic process, diagnosis, prognosis, therapy, and possible diagnostic–therapeutic alternatives, possible risks and complications, and the behaviors they should observe during the observational period. In addition, each patient signed an informed consent, and was advised that some data regarding their therapy would be merged, in an anonymous form, in an experimental clinical research study.

Before the prosthetic rehabilitation, each patient underwent professional oral hygiene and, if needed, routine conservative dentistry or endodontic treatments. All of the patients were motivated and trained to maintain an acceptable oral hygiene condition, and to complete all of the follow-up visits while being alternatively excluded by the study. The prosthetic rehabilitation deemed most suitable and adequate was carried out for each patient, with the cooperation of two expert clinicians, accordingly to the most recent and validated evidence-based dentistry on such topics.

### 2.2. Inclusion Criteria

- Patients with monolithic zirconia single crowns in posterior sectors;
- Crowns screwed on the implant abutment;
- Good oral hygiene;
- No temporomandibular disorders;
- Absence of systemic diseases that contraindicate minor oral surgery.

### 2.3. Exclusion Criteria

- Poor oral hygiene;
- Severe periodontal disease;
- Parafunctions;
- Presence of temporomandibular disorder;
- Systemic disease that contraindicates the surgical placement of implant;
- Patients who do not complete the follow-up period.

### 2.4. Operative Protocol

The clinical protocol was structured as follows:

1. Initial evaluation: anamnesis, panoramic radiograph, and eventual targeted intraoral radiograms, and when necessary, CT or CBCT; preliminary impression by irreversible hydrocolloid impression material (Alginate—Kromopan LASCOD, Sesto Fiorentino, Italy) and analysis of the study models.

2. Surgical procedure: after local anesthesia, a full-thickness flap was elevated, and the implant (Neoss-ProActive Straight, Neoss-ProActive Tapered, Milano, Italy) was inserted according to the actual guidelines about operating protocol and direct closure of the implant site with cover screw application.
3. Exposure of the cover screw and substitution with the healing abutment after 4 months.
4. After 7 days, polyether impression of the implant fixture with individual tray (Impregum™, Saint Paul, MN, USA).
5. The temporary crown of acrylic resin (PMMA) is applied and screwed onto the fixture with a temporary titanium abutment.
6. Positioning of the second PMMA prototype for an appropriate functional and aesthetic evaluation of the emergence profile, with articulation paper use (8-μm aluminum Shimstock, Coltène, Altstätten, Switzerland) to detect premature contacts and interference during centric and eccentric mandibular movements. Aesthetic evaluation in regard to morphology and color using the Easyshade® Compact (VITA® North America, Yorba Linda, CA, USA) spectrophotometer.
7. Positioning the "raw" prosthetic product in 5Y-PSZ Zirconia Biodynamic Zirconium Multilayer 1200/600 Mpa Progressive (Biodynamic, Correggio, Italy) assessing morphological checks and functional adaptation.
8. Bonding on the definitive prosthetic abutment.
9. Final decontamination of the fixture and application of the crown with 40 N of screwing force.
10. Closure of the access hole with Teflon and flow composite resin (Versite Flow Composite, Kerr Dental, Orange, CA, USA).
11. Clinical follow-up visits at 1, 3, 6, and 12 months, and annual check-ups were performed for the following 6 years.

### 2.5. Clinical Evaluation

The same two expert clinicians always conducted the clinical evaluations. A bibliographic search was conducted to select and use the most widely accepted standardized parameters for the clinical assessment; as a result, the authors decided to adopt the modified USPHS criteria (United States Public Health Service criteria), which give standardized clinical results of fixed prostheses on dental implants during the observational period. The USPHS criteria used in this study are reported and listed in Table 1.

**Table 1.** Definition of USPHS criteria from alpha (excellent) to delta (to be replaced).

| USPHS CRITERIA | Alpha (A) | Bravo (B) | Charlie (C) | Delta (D) |
|---|---|---|---|---|
| MARGINAL ADAPTION | No visible crack or penetration of the probe | Visible crack but no penetration of the probe | Visible crack and probe penetration | Restoration must be replaced |
| SURFACE ROUGHNESS | Polished surface, no roughness | Slight roughness, not detectable with saliva film | Moderate roughness, detectable even with saliva film | Restoration must be replaced |
| COLOR MATCHING | No mismatch Perfect integration | Mismatch < 1 grade of Vita shades | Mismatch > 1 grade of Vita shades | Restoration must be replaced |
| ANATOMIC MORPHOLOGY | Natural and physiological anatomy | Unnatural anatomy | Loss of important anatomical structure | Restoration must be replaced |
| FRACTURE | No fracture | Minor fracture (polishable) | Major fracture (not polishable) | Restoration must be replaced |

Overall, this approach allowed the authors to distinguish the prosthetic restorations into four categories as follows:

1.  "EXCELLENT" if the restoration has only "a" score;
2.  "SUCCESS" if the restoration has at least a "b" score;
3.  "MAINTENANCE" if the restoration has at least a "c" score;
4.  "FAILURE" if the restoration has at least a "d" score or the presence of non-correctable adversity.

In addition to the clinical evaluation, the authors collected different technical information regarding the implant localization, the characteristic of the opposite teeth, and the most common adversities (indicating when they occurred). These blank forms are reported in Tables 2 and 3.

**Table 2.** Localization of the prosthetic rehabilitation and characteristic of the opposite tooth.

| LOCALIZATION | | | |
|---|---|---|---|
| REGION | Maxilla | Mandible | |
| TOOTH | Premolar | Molar | |
| SEX | Male | Female | |
| ANTAGONIST ELEMENT | Natural teeth | Prosthetic | Denture |

**Table 3.** Technical adversities during follow-up, specifying if they are reparable or irreparable when they occurred.

| TECHNICAL ADVERSITIES | Reparable | Irreparable |
|---|---|---|
| Debonding crown abutment | | |
| Screw loosening | | |
| Implant failure | | |
| Drop-out | | |

It is relevant to underline that all of the patients were followed-up 3 times per year by a dental hygienist to check the periodontal status and perform the necessary oral hygiene procedures and improve domiciliary hygiene. A success was considered when the SC belonged to the "Excellent" USPHS parameter, keeping the same morpho-functional properties of the first-day evaluation. Survival was considered when the restoration did not need to be replaced. The USPHS scoring, rehabilitation characteristics, and possible adversities were collected into a single database.

A survival treatment was considered when the rehabilitation was not definitely lost or replaced, while success was considered when it maintained excellent USPHS parameters over the 7 years. A Kaplan–Meier plot was used to represent the success rate using Stata® version 13.0 (Stata Corp., College Station, TX, USA).

## 3. Results

In total, 48 patients were included in the study; however, seven did not complete the whole 7 years of follow-up, and thus dropped out of the study. The final sample was 41 patients, 15 males (36.6%) and 26 females (63.4%); as for localization of the prosthetic restorations, 7 (17.1%) were upper premolars, 14 (34.1%) were upper molars, 5 (12.2%) were lower premolars, and 15 (36.6%) were lower molars. Globally, 21 cases (51.2%) were maxillary restorations, 20 (48.8%) mandibular, 29 were molars (70.7%), and 12 were premolars (29.3%). The opposite teeth comprised 31 (75.6%) natural teeth, eight with reconstruction by composite resin, and 10 (24.4%) were prosthetic teeth with ceramic surfaces. The USPHS scores per year are reported in Table 4.

**Table 4.** Distribution of the USPHS scores for the seven years of clinical observation. No C and D parameters were reported.

| USPHS Score Per Years | Marginal Adaption | Surface Roughness | Color Match | Anatomy | Fracture | Total |
|---|---|---|---|---|---|---|
| 1 Year | | | | | | |
| A | 41 | 41 | 41 | 41 | 41 | 41 |
| B | | | | | | 0 |
| 2 Year | | | | | | |
| A | 41 | 41 | 41 | 41 | 41 | 41 |
| B | | | | | | 0 |
| 3 Year | | | | | | |
| A | 41 | 41 | 41 | 41 | 41 | 41 |
| B | | | | | | 0 |
| 4 Year | | | | | | |
| A | 41 | 41 | 41 | 41 | 41 | 41 |
| B | | | | | | 0 |
| 5 Year | | | | | | |
| A | 41 | 39 | 41 | 41 | 41 | 39 |
| B | | 2 | | | | 2 |
| 6 Year | | | | | | |
| A | 41 | 38 | 41 | 41 | 41 | 38 |
| B | | 3 | | | | 3 |
| 7 Year | | | | | | |
| A | 41 | 37 | 40 | 41 | 41 | 36 |
| B | | 4 | 1 | | | 5 |

All 41 patients (100%) showed "excellence" USPHS ratings between the first and fourth years after rehabilitation. In the fifth year of observation, two USPHS criteria changed for surface roughness; so, globally, the USPHS score was "excellence" in 39 cases (95.1%), with only 2 (4.9%) considered as "success". In the sixth year, one more adversity was added to a patient as showing surface roughness classified as "b"; thus, globally, 38 cases (92.7%) were "excellent" and 3 (7.3%) were "success" after 6 years of observation. During the last (seventh) year, two other patients were classified with "b", precisely one for increased surface roughness and one for a minor color mismatch. Finally, the USPHS score was "excellent" in 36 (87.8%) cases and "success" in 5 (12.2%) at the seventh year of clinical evaluation. The trend of the USPHS score is reported in Figure 1.

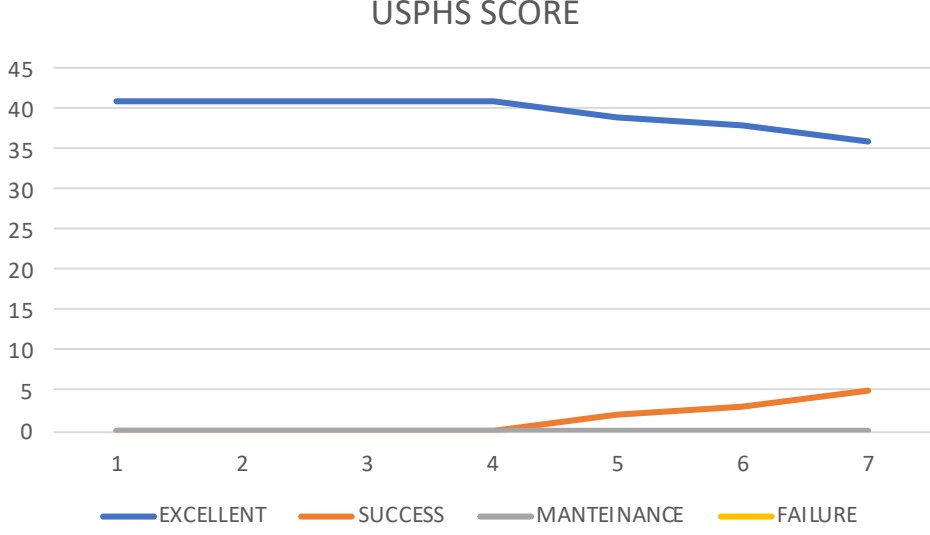

**Figure 1.** USPHS score trend from the first to the seventh year of clinical evaluation.

Different technical adversities were reported over seven years: screw loosenings in three patients at 64, 70, and 76 months, which were tightened. Moreover, four composites filling opposite teeth were fractured and reconstructed adequately with a composite overlay. Overall, no catastrophic complications were registered (Figure 2).

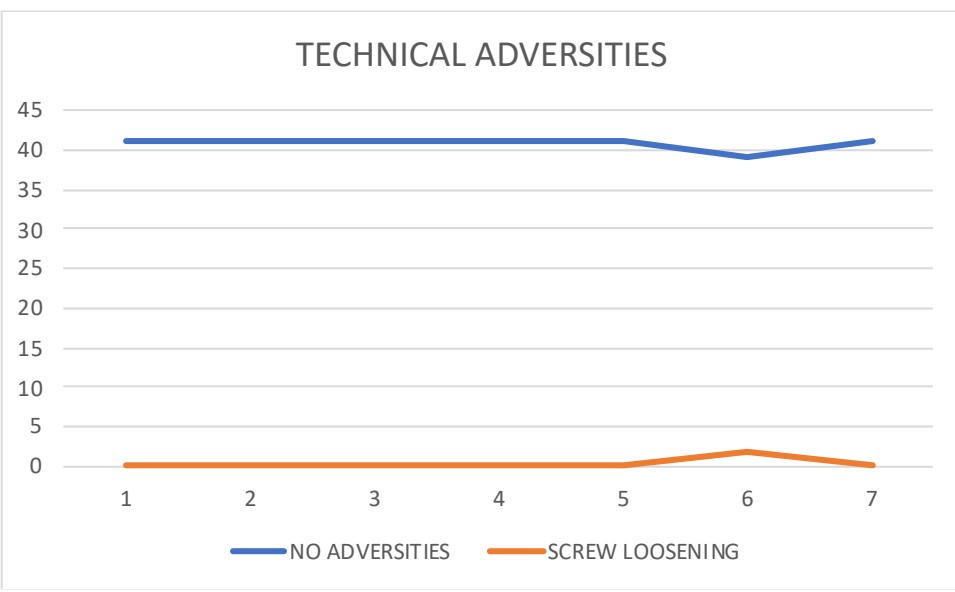

**Figure 2.** Trends in numbers and types of technical adversities reported during the seven years of clinical evaluation.

The cumulative survival rate was 100%, and the success rate was 87.8% (Figure 3).

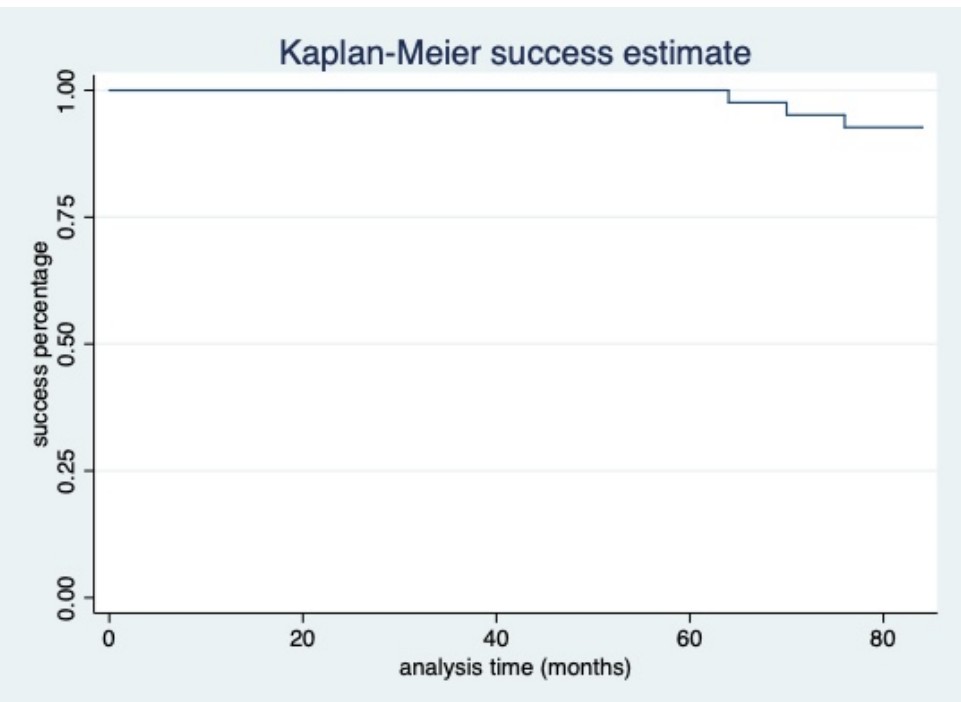

**Figure 3.** Kaplan–Meier success plot demonstrates a good outcome for zirconia rehabilitation after 7 years.

## 4. Discussion

The present study aimed to assess the clinical properties of monolithic zirconia SC screwed-on implants, comparing the obtained results to those of other similar studies that considered the clinical evaluation with USPHS criteria.

The localizations of the restored teeth were homogeneous: 24 were molars, and 24 were premolars with different occlusal loads between them, and the 24.4% of antagonists were represented by ceramic surfaces of fixed prostheses in order to reproduce the common situation of daily practice, which is different from Weigl et al., who decided to consider exclusively natural teeth as the antagonists [18].

The clinical performances evaluated in such a long follow-up period led to an 88.8% overall success rate, which could be considered a good outcome achieved among our patients. Our success results were compared with those of other authors that followed USPHS evaluation criteria, and are consistent with Kraus's research, and are quite identical to the 88.2% success rate of Degidi [9,19], even considering that Kraus exclusively evaluated anterior teeth and Degidi considered three-unit fixed dental prostheses too. These results differed from those of Spies et al., who reported a 38.5% success rate for veneered zirconia, a low value mainly due to veneer chipping [20]. Regarding the survival rate, the authors reported 100% survival, which is consistent with that of several authors and superior to Korsch, although our study referred to different follow-up times [6,19,21–23].

The authors focused on clinical USPHS parameters, comparing them with the present literature. It is important to consider that although there are too many studies conducted on such kinds of restorations, there is a lack of studies that evaluate monolithic zirconia with standardized USPHS parameters, exclusively the single-crown screwed-on implants are evaluated; thus, it was mandatory to compare our results to studies conducted on FDP or veneered zirconia. The studies that evaluated that specific population of prosthetic restoration have short-term follow-ups, with a maximum of three years [14,24,25]. In the first year of evaluation, each patient showed excellent parameters (41 on 41 alpha), which have to be considered better outcomes than the 1-year follow-up study of Pol et al., who reported bravo percentages of 21.4% and 3.6% regarding the marginal adaption and the opening of the anatomical morphology, respectively; however, they reported 100% survival, which is consistent with this study [26]. The clinical situation remained stable through the years until the fourth year. This is in contrast with a similar study of Spies et al., which essentially reported progressive deterioration regarding surface roughness and marginal integrity since the second year [20]. In addition to the clinical parameters, the survival rate reported was 100% at 5 years, which is consistent with this study. In the last 3 years of observation, a worsening of clinical parameters was reported. In fact, in the fifth year, two patients showed an increase in surface roughness. This switch from Excellence to Success of this specific USPHS parameter was reported in two more patients, one in the sixth year and the last in the seventh year of evaluation. In every case, it was a slight entity that was resolved by polishing the surface, as suggested by Kheur et al. [27]. The worsening of surface roughness is consistent with that found in previous studies [12,19,20], so it deserves to be studied further.

It is interesting to report that the intraoral grinding with diamond burst was performed in all of these cases to achieve occlusal adjustments. This result can be explained by the LTD, also known as "aging", which occurs when the zirconia crystalline form changes from tetragonal to monoclinic, and is due to water penetration into the zirconia surface via microcracks opened by the diamond burst [28], also increasing bacterial biofilm adhesion [29]. In fact, authors, according to the literature, suggest that it is mandatory to limit, only when necessary, intraoral occlusal adjustments to avoid the development of these biological and technical adversities [30].

The last patient who was downgraded from Excellence to Success showed that a discoloration of restoration occurred during the last year of observation, representing 2% of the entire sample. This value is in contrast with the 10% observed by Degidi, and the 13.7% and 15% reported by Worni and Muhlemann, respectively [9,21,31]. More-

over, in this single case, the framework was subjected to occlusal adjustment, confirming Herpel et al.'s hypothesis that discoloration could also be attributed to an excessive trimming from the definitive prosthetic framework [32].

The marginal adaption was correct and stable during the observational period in 100% of the cases, similar to results reported by Worni [21]. Probably, worsening of the marginal adaption may depend on osteointegration values; if implants have no bacterial infiltration or mucositis or peri-implantitis, marginal adaptation remains stable and is further favored by the excellent biocompatibility of zirconia and perfect control of the periodontal status [33]. Globally, the success and survival rates obtained in this long-term follow-up study are consistent with the fourteen-year follow-up of the study conducted by Zarone et al. on FDP, which reports an overall survival rate of 98%, and a success rate of 83% [16]. The only failure in their study was a catastrophic fracture; however, they suggest zirconia as a viable prosthetic solution for implant-supported rehabilitations.

In addition to the good clinical properties analyzed before, technical adversities must be considered to carry on a global evaluation of this type of restoration to suggest it in daily practice. Chipping, the most common adversity reported in metal–ceramic and zirconia-veneered implants, was impossible because monolithic zirconia does not present feldspathic veneers. In the current study, screw loosening was the only technical problem, and occurred in three cases (7.3%) between the fifth and seventh years of follow-up, as observed in other long-term studies [10,34].

No fracture of the screw, abutment, or framework occurred, and this could undoubtedly be related to the exclusion of parafunctional patients, but also to accurate control of the occlusal load; in fact, the high occlusal load is generally considered as a risk factor for increasing the occurrence of technical complications such as screw loosening, screw and abutment fracture, and implant failure, especially if the rehabilitation is composed of monolithic zirconia, which has a low elasticity modulus [12]. Among our cases, no biological adversities were registered, as no implant failure occurred in 7 years of follow-up, thus remaining consistent with the results of Worni, Krause, and Kolgeci [19,21,34]. These data further stress the importance of maintaining excellent periodontal health in implant dentistry, regardless of the type of prosthetic rehabilitation.

No opposite wearing was registered, because prosthetic protocols were followed entirely using the first PMMA provisional to define soft tissues and the second PMMA prototype of a definitive crown to establish the final crown anatomy. Moscovitch also described a similar protocol, and generally consented to perform occlusal adjustments directly on the provisionals, hence avoiding trimming the definitive crown [35]. Moreover, a perfectly polished surface reduces antagonist wearing, making it comparable to normal teeth [21]. Furthermore, the latest research suggests that the addition of nanoparticles in PMMA provisionals could reduce bacterial and mycotic adhesions, resulting in a better soft tissue response [29,36].

The last technical topic that is mandatory to consider involves the advantages and disadvantages of screwed prostheses compared to cemented-on implants. The data from the literature about the frequent technical complications (mainly screw loosening, screw fracture or framework fracture) are still contrasting [13]. There are mainly two disadvantages of screw-retained prostheses: the screw hole that could be located in the aesthetic zone of the crown (depending on implant insertion axe), and loosening of the screw that connects the crown to the abutment. In this regard, Assenza et al., in their interesting study, showed an increase in the vascular endothelial growth factor (VEGF) concentration in peri-implant tissues of un-screwed abutments, demonstrating that peri-implant tissue inflammation may occur due to bacterial penetration after screw loosening [37]. The main advantage of the screw-retained prosthesis is the retrievability in every technical adversity that could occur, such as the abutment loosening [38]. Conversely, cement-retained rehabilitation overcomes aesthetic (presenting no screw hole) challenges and screw loosening, but may lead to biological problems due to the frequent occurrence of cement surpluses that could remain in periodontal spaces, leading to marginal tissue inflammation, marginal adapta-

tion worsening, bone loss, peri-implantitis, and potential implant failure [38]. As for such considerations, the authors focused exclusively on screw-retained crown rehabilitations, in order to better understand the advantages and disadvantages of this subpopulation of fixed prosthetic restorations over such a long period.

This study has some limitations: the small and convenient sample, due to the difficulties of following many patients over seven years; the lack of controls; the individual unblinded clinical evaluation; and the lack of statistical analysis can be considered weakness points. Moreover, the satisfaction of the patients was not considered. Further RCT studies are possible with prolonged follow-up, and a correct statistical analysis needs to be conducted to confirm or potentially improve our results. Currently, the monolithic zirconia represents a valid alternative to the metal–ceramic crown, and, despite its slight optical problems, it is a predictable material that could be used in fixed posterior rehabilitation supported by implants thanks to its excellent biological behavior and mechanical properties. This long-term clinical study confirms the hypothesis that monolithic zirconia as a single-crown screw-retained implant could represent a viable prosthetic solution.

Clinical case (Figure 4).

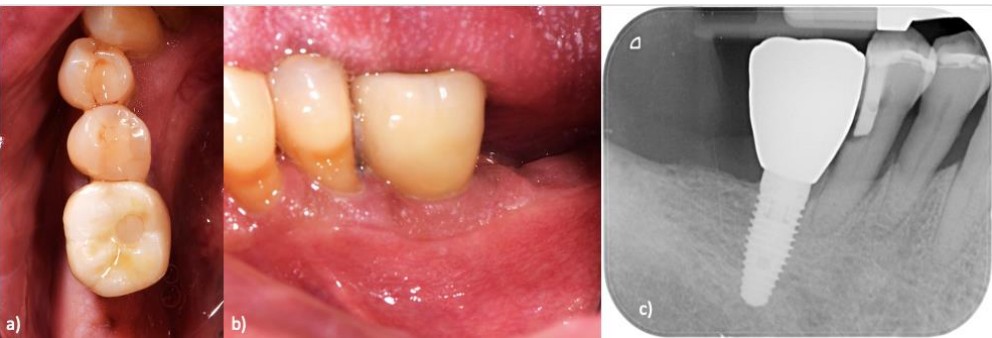

**Figure 4.** Single-crown monolithic zirconia, screwed-on implant, for replacement of a missing first inferior molar: (**a**) occlusal view; (**b**) lateral view; (**c**) radiological appearance after 7 years.

**Author Contributions:** Conceptualization, G.B. and M.C.; methodology, G.B.; validation, S.C.; formal analysis, S.C.; investigation, A.V. and G.M.; resources, G.M.; data curation, G.D.R.; writing—original draft preparation, G.B.; writing—review and editing, M.C. and S.C.; visualization, M.C.; supervision, M.C. All authors have read and agreed to the published version of the manuscript.

**Funding:** This research received no external funding.

**Institutional Review Board Statement:** The study was conducted in accordance with the Declaration of Helsinki, and the protocol was approved by the Ethical Committee of University Hospital "Policlinico di Bari" (Study No. 7390, N. Prot. 0069684).

**Informed Consent Statement:** Written informed consent was obtained from the patients to publish this study. As provided in the specific section of the Italian Code of Medical Ethics 'Information and communication. Consent and dissent—Title VI', each patient was provided with comprehensible and comprehensive information on the techniques and materials used, on the diagnostic process, on the diagnosis, on the prognosis, on the therapy, and on any diagnostic–therapeutic alternatives, on the possible risks and complications, as well as on the behaviors that they should observe during the whole treatment. In addition, each patient signed an informed consent, and was advised that some data regarding their therapy would be merged, in anonymous form, in an experimental clinical research study.

**Data Availability Statement:** Not applicable.

**Conflicts of Interest:** The authors declare no conflict of interest.

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
