# Peer review of "Screwed Monolithic Zirconia Crowns for Mono-Implant Posterior Rehabilitation: A Prospective Clinical Study on 41 Patients with a 7-Year Follow-Up"

_prosthesis, doi:10.3390/prosthesis5040072_

Round 1

Reviewer 1 Report

The article entitled "SCREWED MONOLITHIC ZIRCONIA CROWNS FOR MONO-IMPLANT POSTERIOR REHABILITATION: A PERSPECTIVE CLINICAL STUDY ON 41 PATIENTS WITH 7 YEARS FOLLOW-UP" aims to evaluate the qualitative status of implant-supported screw-retained monolithic single crowns (SC). The article is well written, the material and methods used are appropriate. The results and discussion are well reported, so the article can be published in its present form.

Author Response

Dear reviewer,

thank you very much for the appreciation words about our work. 

Best regards.

Reviewer 2 Report

Abstract: Needs extended editing (Veneer and monolithic zirconia are not discussed in introduction). Sentences are not bonded to each other

Introduction: Unnecessary classification which is well known to dentists I suggest to delete(46-53).

Ref 5 authors??

I suggest the classification that is recommended in:

Zhang Y, Lawn BR. Novel Zirconia Materials in Dentistry. J Dent Res. 2018;97(2):140-147. or an older Tzanakakis, E.; Tzoutzas, I.; Kontonasaki, E. Zirconia: Contemporary views of a much talked material: Structure, applications and clinical considerations. Zirconia Hel. Stom. Rev. 2013, 57, 101–137. 

lines 55-68 also common knowledge please condense

line 76 : ...since from...please rephrase and get an extended english editing for major grammar mistakes.

M&M are OK and clinical evaluation OK

Tables and Figures should have better analysis

Discussion: The whole structure requires better flow starting from the analysis of the results . I think this part should be re-written.

Conclusions are OK.

Needs extended English editing. Essential grammar mistakes underestimate the overall well organized clinical trial.

Author Response

Dear reviewer,

authors want to thank you for such constructive comments which surely improve the scientific soundness and the readability of our paper. 

Here is a point-by-point address to your comments.

Abstract: Needs extended editing (Veneer and monolithic zirconia are not discussed in introduction). Sentences are not bonded to each other

As your right suggestion, abstract was not correct, so was quite rewritten, removing the unnecessary notions and improving the fluency. Thank you

Introduction: Unnecessary classification which is well known to dentists I suggest to delete(46-53).

This classification was real unnecessary and was deleted. Thank you for this suggestion.

Ref 5 authors??

Old ref 5 (now ref 6) was fixed in the bibliography thank you.

I suggest the classification that is recommended in:

Zhang Y, Lawn BR. Novel Zirconia Materials in Dentistry. J Dent Res. 2018;97(2):140-147. or an older Tzanakakis, E.; Tzoutzas, I.; Kontonasaki, E. Zirconia: Contemporary views of a much talked material: Structure, applications and clinical considerations. Zirconia Hel. Stom. Rev. 2013, 57, 101–137. 

lines 55-68 also common knowledge please condense

Lines 55-68 were condensed and, as your suggesting, was added the classification of Zhang et al., which is more updated and more adequate to the context. Thank you very much about this.

line 76 : ...since from...please rephrase and get an extended english editing for major grammar mistakes.

English was edited, now it could sound better. Thank you

M&M are OK and clinical evaluation OK

Tables and Figures should have better analysis

Tables and Figures were completed with more complete captions. Thank you 

Discussion: The whole structure requires better flow starting from the analysis of the results . I think this part should be re-written.

The whole discussion was deeply edited. Reporting firstly the results analysis resulted in an improved readability with more logical sequence of the sentences.

English was subjected to an extensive editing, especially regarding grammars and we want to thank you for your comments that reflect the absolute mastery and knowledge of a reviewer expert in this field. As said before, this surely contributes to improve the quality of our work. 

Conclusions are OK.

Round 2

Reviewer 2 Report

Ref #14 and 48 should be checked

Much improved overall

Author Response

Dear reviewer,

Thanks for your gentle words. The ref. n. 14 and 28 were removed. 

Thank you very much on behalf of all the authors.